# Combination of Anti-Angiogenics and Immunotherapies in Renal Cell Carcinoma Show Their Limits: Targeting Fibrosis to Break through the Glass Ceiling?

**DOI:** 10.3390/biomedicines12020385

**Published:** 2024-02-07

**Authors:** Manon Teisseire, Sandy Giuliano, Gilles Pagès

**Affiliations:** University Cote d’Azur (UCA), Institute for Research on Cancer and Aging of Nice, CNRS UMR 7284; INSERM U1081, Centre Antoine Lacassagne, 06189 Nice, France; manon.teisseire@etu.univ-cotedazur.fr

**Keywords:** renal cell carcinoma, TKI, immunotherapies, resistance, fibrosis, CTGF

## Abstract

This review explores treating metastatic clear cell renal cell carcinoma (ccRCC) through current therapeutic modalities—anti-angiogenic therapies and immunotherapies. While these approaches represent the forefront, their limitations and variable patient responses highlight the need to comprehend underlying resistance mechanisms. We specifically investigate the role of fibrosis, prevalent in chronic kidney disease, influencing tumour growth and treatment resistance. Our focus extends to unravelling the intricate interplay between fibrosis, immunotherapy resistance, and the tumour microenvironment for effective therapy development. The analysis centres on connective tissue growth factor (CTGF), revealing its multifaceted role in ccRCC—promoting fibrosis, angiogenesis, and cancer progression. We discuss the potential of targeting CTGF to address the problem of fibrosis in ccRCC. Emphasising the crucial relationship between fibrosis and the immune system in ccRCC, we propose that targeting CTGF holds promise for overcoming obstacles to cancer treatment. However, we recognise that an in-depth understanding of the mechanisms and potential limitations is imperative and, therefore, advocate for further research. This is an essential prerequisite for the successful integration of CTGF-targeted therapies into the clinical landscape.

## 1. Introduction

The prognosis of renal cell carcinoma has improved in recent years thanks to the clinical application of various therapies, including, above all, anti-angiogenesis therapy and, more recently, immune checkpoint inhibitor therapy. Nevertheless, renal cell carcinoma cannot be cured, and there is still room for improvement through deciphering the molecular mechanisms leading to resistance to current treatment and discovering new therapeutic targets. This review summarises the latest findings in the field and highlights one of the key players, CTGF, which is involved in fibrosis, an important feature of precancer.

## 2. Clear Cell Renal Cell Carcinoma

Renal cancer comprises a relatively small portion, 2–5%, of global cancer diagnoses, impacting around 400,000 individuals. The predominant histological subtype is clear cell renal cell carcinoma ccRCC (ccRCC), which accounts for 80% of cases, followed by papillary RCC (10–15%) and chromophobe RCC (5%) [1]. Each subtype has different histological and genetic features, which affect treatment methods and prognosis. For localised RCC, partial or total nephrectomy has been shown to be effective [2]. However, 25% of patients are diagnosed at the metastatic stage, necessitating systemic treatments. Unfortunately, metastatic renal cell carcinoma has a bad prognosis, with a five-year survival rate of only 10–15% [3]. 

## 3. Anti-Angiogenic Therapies: Beneficial Effects and Limitations

Among the various renal cell carcinomas, ccRCC, the most common form, is probably one of the most vascularised tumours. The recurrent inactivation of the von Hippel Lindau gene, which leads to the stabilisation of Hypoxia Inducible Factor 1 and 2 α (HIF1/2α), is one of the causes of this hypervascularisation. VHL inactivation is key for the expression of one of the most important pro-angiogenic factors, vascular endothelial growth factor (VEGF). Therefore, ccRCC represents a paradigm for treatment with anti-angiogenic drugs [4], including the historical combination of bevacizumab/Avastin (humanised monoclonal antibody) with interferon-alpha [5] and the use of the multitarget inhibitor sunitinib [6]. More recently, therapies approved between 2007 and 2009 have been gradually replaced by more potent therapies targeting alternative tyrosine kinase receptors (PDGFR, c-MET, FGFR) implicated in mechanisms of resistance, including axitinib [7], pazopanib [8], tivozanib [9], cabozantinib [10], and lenvatinib [11]. Despite their relative effectiveness, these treatments are not a curative solution and patients progress after varying lengths of time. The heterogeneous responses of patients can be divided into three distinct categories: (1) patients with intrinsic resistance who exhibit no or minimal response, facing rapid progression and mortality; (2) patients who have acquired resistance and experience transient benefits followed by relapse; (3) and a subgroup for whom treatment proves effective over an extended timeframe [12]. This diversity of patient outcomes underscores the multifaceted nature of resistance, which is due to factors such as genetic or epigenetic aberrations in cancer cells, environmental influences, and interactions with host cells. These mechanisms can either manifest themselves before the start of therapy or occur later. Each patient has a unique genetic profile, which requires a personalised therapeutic approach. However, achieving this goal requires a comprehensive understanding of the specific genetic landscape and the underlying mechanisms that determine adaptation to treatments with varying efficacy. Resistance identified in cancer therapies can be divided into intrinsic and acquired forms depending on the temporal context of its manifestation. This review examines in detail the temporal dimensions that influence resistance and provides insights into the different mechanisms underlying this phenomenon (see Figure 1). In addition, possible ways of deciphering the factors contributing to treatment failure are outlined.

## 4. Innate Anti-Angiogenic Resistance Leads to Primary Drug Inefficacy

### 4.1. Molecular Subclassification of ccRCC

As outlined by Beuselinck B. et al. [13], ccRCC manifests in four distinctive molecular subtypes, labelled ccRCC1 to 4. Among these, ccRCC2 (“classical”) and ccRCC3 (“normal-like”) exhibit the loss of tumour suppressor genes PBMR1 and VHL, resulting in a robust activation of proangiogenic pathways. Notably, these subtypes display heightened sensitivity to anti-VEGF/VEGFR interventions. Furthermore, ccRCC1 (“c-myc up”) and ccRCC4 (“c-myc-up and immune up”) are characterised by low tumour cell differentiation, the absence of the tumour suppressor gene BAP1, diminished expression of pro-angiogenic genes, and PDL-1 overexpression. Consequently, these subtypes demonstrate reduced responsiveness to anti-VEGF/VEGFR treatments but heightened sensitivity to immune checkpoint inhibitors (ICI) compared to their counterparts. It is worth noting that subtypes of ccRCC1 and 4 exhibit a shorter overall survival in contrast to ccRCC2 and 3. 

### 4.2. Single Nucleotide Polymorphism (SNPs)

Germline genetic alterations in the promoter region of different genes have been shown to be clinically relevant and are potential predictive markers for the efficacy of anti-angiogenic therapies. SNPs, which are characterised by subtle genetic changes in which a single nucleotide is exchanged, exert a significant influence. SNPs can predict response to treatment, susceptibility to environmental factors, and predisposition to certain diseases. In ccRCC, SNPs play a central role as they influence both development and resistance to anti-angiogenic therapies. The VEGF gene, for example, is highly polymorphic, and many single nucleotide polymorphisms (SNPs) have been described [14]. Furthermore, a compelling illustration of this phenomenon is found in the intriguing correlation between specific IL-8 SNPs and reduced survival rates under sunitinib treatment. This correlation is a”consequence of the activation of alternative angiogenic pathways, underscoring the profound implications of SNPs in shaping the dynamics of response to therapeutic interventions [14]. 

### 4.3. miRNAs

Numerous miRNAs have been implicated in orchestrating TKI resistance in ccRCC by targeting crucial signalling pathways that govern cell proliferation, survival, and drug response. Operating at the post-transcriptional level, these miRNAs exert a negative influence on gene expression. Within the landscape of ccRCC, specific miRNAs have surfaced as both prognostic markers [15] and predictive indicators for sunitinib response [15,16]. miR-15b was found to be significantly upregulated in cell lines resistant to sunitinib after exposure to the drug. This observation has been replicated in in vivo models [17]. Other notable miRNAs overexpressed in sunitinib-resistant RCC cells include miRNA-575, miRNA-642b-3p, and miRNA-4430 (in vitro study), as well as miRNA-942, miRNA-133a, miRNA-628-5p, and miRNA-484 (in vivo study) [16,18]. Overexpression of miR-144-3p in ccRCC was shown to increase cell proliferation, clonogenicity, migration, invasion, and resistance by inhibiting the expression of the tumour suppressor gene ARID1A, which encodes the basic directional subunit of SWI/SNF chromatin remodelling complexes [19]. In addition, miR-221 attenuates the efficacy of sunitinib by reducing VEGFR2 expression. Additionally, miR-942 up-regulates MMP-9 and VEGF secretion, which promote endothelial cell migration and sunitinib resistance [16]. In contrast, miR-200b and miR-141 were found to be downregulated in ccRCC compared to benign tissue, and their expression may serve as an independent prognostic factor for prolonged progression-free survival and overall survival [20].

Immediate/rapid resistance:
Molecular subclasses of mRCCs;miRNAs and SNPs;Lysosomal sequestration of sunitinib;Metabolic adaptations.Early resistance:Communication between tumour cells.Late resistance:
c-MET-dependent EMT;Communication with the microenvironment: vascular and lymphatic endothelial cells, CAFs, TAMs, MDSCs.

## 5. Tumour Cells Adaptative Response after Anti-Angiogenic Treatments

As we have previously published, resistance to anti-angiogenic drugs that mainly target the VEGFA/VEGFR axis is time-dependent [17]. Originally, the mechanism of action of these drugs was defined as inhibitors of vascular network development. The elegant hypothesis proposed that by inhibiting tumour vascularisation, these drugs should eradicate the tumour by asphyxiating it and limiting nutrient access. In this way, these treatments should not have induced genetic adaptation of tumour cells, whose genetic plasticity is the origin of resistance. 

Tumour cells also express receptors or cytokines targeted by these therapeutics. In addition, chronic exposure to so-called anti-angiogenic drugs leads to an adaptation of the tumour cells. This adaptation process includes genetic changes, a reduction in drug delivery to their targets and the production of excessive angiogenic cytokines. For example, other members of the VEGF family and angiopoietins, together with inflammatory cytokines, also have pro-angiogenic properties.

### 5.1. Lysosomal Sequestration

An immediate adaptation to TKIs, including the longstanding standard treatment sunitinib, involves sequestration within acidic intracellular compartments, specifically the lysosomes. Lipophilic drugs (logP > 2) with ionisable amines (pKa > 6) accumulate in lysosomes by passive diffusion and/or actively by efflux ABC pumps, where they are protonated, can no longer diffuse back through the membrane and are sequestered. Autophagy and lysosomal metabolic pathways serve to degrade cytoplasmic materials and organelles. Lysosomes are partners in the degradation process of autophagy. The alteration of their pH by lysosomotropic drugs, such as sunitinib, inhibits autophagic processes [21,22]. This intracellular shielding effectively hinders sunitinib’s access to the tyrosine kinase domain of target receptors in the cytoplasm, thereby limiting its therapeutic activity. Remarkably, this sequestration phenomenon is observed in both tumour and endothelial cells [21,23]. Sunitinib instigates incomplete autophagy while concurrently stimulating the expression of pro-inflammatory cytokines and the ABCB1 transporter. In the absence of the autophagy degradative pathway, a dependence on the proteasome for the elimination of unfolded proteins is observed. Consequently, the inhibition of lysosome trapping by LLOMe (L-Leucyl-L-leucine methyl ester), the suppression of the ABCB1 transporter by elacridar, or the hindrance of the proteasome by bortezomib re-sensitise cells to the therapeutic effects of sunitinib [21].

### 5.2. EGFR Mutations

By exerting a pressure of selection on tumour cells, anti-angiogenic drugs induced genetic adaptations within a few hours, leading to a modification in gene expression. One striking modification is the decreased expression of phosphotyrosine phosphatase receptor kappa, resulting in the constitutive activation of EGFR [24]. Such constitutive activation renders the tumours more sensitive to EGFR inhibitors, especially those used for the treatment of lung cancer.

### 5.3. Methylation of Tumour Suppressor Genes

Enhancer of Zeste Homolog 2 (EZH2) stands as the catalytic core component of the Polycomb Repressive Complex 2 (PRC2), orchestrating gene silencing through the trimethylation of H3K27, thereby inducing transcriptional repression. The dysregulation of EZH2-mediated methylation emerges as a significant factor in conferring resistance to tyrosine kinase inhibitors (TKIs).

EZH2 inhibitors increased the efficacy of the EGFR inhibitor gefitinib in cells that do not have an EGFR mutation and are resistant to this TKI [25]. In addition, EZH2 has a negative correlation with MET, a key player in resistance to EGFR TKIs in non-small cell lung cancer (NSCLC) [26]. In ccRCC, the overexpression of EZH2 has been documented and is intricately associated with a poor prognosis [26,27]. Overexpression of EZH2 induces methylation of the LATS1 promoter and suppresses the expression of this tumour suppressor [28]. LATS1 is a key component of the Hippo signalling pathway. When the Hippo signalling pathway is activated, LATS1 induces the degradation of YAP/TAZ, thereby maintaining a state in which the growth and metastasis of cancer cells are inhibited [29]. Deciphering the role of EZH2 in methylation processes sheds light on its potential as a key player in mediating resistance mechanisms and provides valuable insights for targeted therapeutic interventions [30].

### 5.4. Secretory Phenotype

More importantly, at longer time points after the initiation of anti-angiogenic therapies, tumour cells modified their secretome to shape the tumour microenvironment. One of the first hypotheses of resistance to anti-angiogenic drugs was the presence of alternative angiogenic factors redundant to VEGF. These factors may be present before treatment or may be involved in the processes of adaptation to treatment. Among them, the family of ELR+CXCL cytokines (CXCL1, 2, 3, 5, 6, 7, 8), whose founding member is interleukin 8 or CXCL8, exerts both pro-angiogenic and pro-inflammatory processes. These cytokines exert their effects via G-protein-coupled receptors, CXCR1/2 [31]. Their activation stimulates protein kinases C, phospholipase C, and the signalling pathways PI3K/AKT/mTOR, RAS/RAF/MEK/ERK, and NFκB. Activation of the ELR+CXCL/CXCR pathway (i) stimulates tumour cell survival and proliferation, (ii) shapes the microenvironment, and (iii) promotes the metastatic process. Sunitinib, the reference treatment for renal cell carcinoma, triggers an inflammatory response following an incomplete autophagy mechanism. Acute or chronic treatment stimulates the expression of various pro-inflammatory factors (cytokines, chemokines, and growth factors). The expression of CXCL5, an ELR+CXCL cytokine, increases in resistant cells, in contrast to cells undergoing short-term sunitinib treatment. More specifically, CXCL5 is a predictive marker of sunitinib efficacy [32]. The ELR+CXCL/CXCR1/2 axis thus exerts multiple effects, leading to pro-tumour effects [33] and the aggressiveness of ccRCC [31].

## 6. Shaping of the Tumour Microenvironment: Stromal Cells, Immune Cells, and Vessels

### 6.1. Alternative Vascular and Lymphatic Networks

Because of anti-angiogenic therapies, the emergence of alternative vascular networks represents a distinctive way to sculpt a pro-tumour environment. While VEGFA stands out as the most recognised angiostimulatory protein, it is imperative to recognise that angiogenesis can be instigated by a myriad of growth factors. These encompass angiopoietins, epidermal growth factors, fibroblast growth factors, hepatocyte growth factors, transforming growth factors, placental growth factors, and stromal cell-derived factor 1.

Therefore, the complex nature of angiogenesis involves a network of diverse growth factors beyond VEGFA, highlighting the complexity of the processes shaping the tumour microenvironment after anti-angiogenic interventions. Understanding this intricate interplay is crucial for developing targeted strategies that encompass the broader spectrum of angiogenic mediators to optimise therapeutic outcomes.

In this context, lymphatic networks assume a pivotal role as a major facilitator enabling tumour growth and dissemination throughout the organism in advanced tumours [34]. This intricate involvement of lymphatic pathways contributes significantly to the formation of new metastatic sites, emphasising the crucial role played by these networks in the metastatic cascade. Understanding and targeting the dynamics of lymphatic involvement is imperative to build comprehensive strategies to impede tumour progression and metastatic spread [35,36]. However, in low-grade tumours, lymphatic vessels are the primary way to initiate an anti-tumour immune response in the lymph node by activating T cells. But when immune cells are exhausted, the lymphatic network contributes to the spread of tumour cells in addition to metastatic dissemination through the vascular network. 

One of the most important drivers of lymphatic development is VEGFC, which exerts a pro- or anti-tumour role depending on the tumour stage [37]. Experimentally, VEGFC knockout cells fail to induce the development of experimental ccRCC in immunodeficient mice, a context that mimics immunotolerance, whereas they induce a rapidly growing tumour in an immunocompetent context. In this context, anti-VEGFC antibodies also have important anti-tumour activity as they target both tumour cells and the lymphatic network [38]. These results underline the ambivalent role of the lymphatic vessels and their drivers in the tumour context, playing the role of the “good or bad cop”.

Neuropilins (NRP 1 and 2), coVEGF receptors, are also involved in the lymph phangiogenesis of tumours. NRP act as receptors for semaphorins and interact with VEGFs, especially VEGF-C and VEGF-D. RCC cells highly express the VEGF coreceptors neuropilin 1 and 2 (NRP1/2) [39,40]. Therefore, NRP inhibitors have potent anti-tumour effects on experimental RCC in immunodeficient or immunocompetent mouse models. Considering NRPs as immune checkpoints may explain the efficacy of NRP inhibitors targeting tumour cells, endothelial cells, and exhausted immune cells.

### 6.2. Role of Microenvironmental Cells in RCC Aggressiveness

ccRCC primary tumours and metastases are composed of approximately 60% tumour cells, 10% endothelial cells, 9% tumour-associated neutrophils (TAN), 8% macrophages (including myeloid-derived-suppressor cells (MDSC)), 7% T lymphocytes (all types), 3% cancer-associated fibroblasts (CAF), 1% B lymphocytes, 1% dendritic cells, and 1% “natural killer” (data from TCIA (https://tcia.at/ (accessed on 4 December 2023))). The tumour microenvironment is rich in cytokines. They are secreted by tumour or stromal cells (leukocytes, endothelial cells, and fibroblasts) [41] (Figure 2).

RCC tumour cells express several tyrosine kinase receptors, including PDGFR, CSF1R, and c-MET, which are targets of anti-angiogenic drugs such as sunitinib, sorafenib, pazopanib, tivozanib, lenvatinib, and cabozantinib. They also express the VEGF coreceptor neuropilins (NRP1/2), which can be targeted by specific drugs [42]. VEGFR is also expressed on T cells, such as NRPs, and its stimulation leads to T cell exhaustion. In addition, RCC cells express PDL1 and PDL2, TIM3, and CD80/86, which can induce immune tolerance by stimulating PD1, galectin 9, and CTLA4, respectively. In addition to their ability to inhibit tumour angiogenesis, anti-angiogenic drugs, therefore, contribute to the reactivation of the immune system.

#### 6.2.1. Tumour Cells

Tumour cells can influence the tumour microenvironment and, in particular, the immune system through surface proteins and the release of specific chemokines.

Immune tolerance is induced by three major families of surface proteins: (i) the proteins that bind CTLA4: B7-1 (CD80) and B7-2 (CD86), (ii) the proteins that bind PD1: PDL1 and PDL2, and (iii) the TIM3 binding protein: Gal-9 (Galectin-9). These proteins expressed by tumour cells bind to their co-receptors expressed by T lymphocytes and natural killer (NK) cells. This binding depletes lymphocytes and NK and promotes the differentiation of effector T lymphocytes into regulatory T lymphocytes (Treg) [43]. In RCC, immunosuppression induced by the expression of PDL1, PDL2, and B7-1 or B7-2 is inhibited by nivolumab (anti-PD1) or ipilimumab (anti-CTLA4) (treatment referred to as “immune checkpoint inhibitor therapy”). However, other immunotolerant proteins (TIM3, LAG3, etc.) are involved in resistance to immunotherapy. 

Tumour cells overexpress CXCR1/2 and CXCL1/5/7/8. Elevated levels of CXCL7/8 and CXCR2 within the tumour correlate with poor prognosis in non-metastatic patients [44]. The use of an anti-CXCL7 antibody blocks the growth of experimental RCC in immunocompromised mice [44] and CXCL5 stimulates angiogenesis and promotes tumour development and resistance to sunitinib [32]. Moreover, CXCL5 and CXCL7 correlate with a poor response to sunitinib [32,45].

#### 6.2.2. Vascular and Lymphatic Endothelial Cells

Pathological angiogenesis is often associated with resistance to treatment, including immunotherapy. Tumour cells induce the formation of non-functioning blood vessels. Normalisation of blood vessels using anti-angiogenesis drugs promotes access of anti-tumour immunotherapies and immune cells to tumours. Numerous clinical studies, therefore, show better efficacy of immunotherapy in combination with an anti-angiogenic agent. Endothelial cells physiologically express CXCR1/2, the stimulation of which activates pro-tumour angiogenesis, a key phenomenon in the development of ccRCC [46]. This activation compensates for the inhibition of VEGF-dependent tumour vascularisation (by anti-angiogenic agents, inhibitors of VEGFR receptors). Moreover, lymphatic and vascular endothelial cells express the Programmed-Death ligand 1 (PDL1), which restricts the T cell response [47,48] and trans-endothelial cell migration [49].

#### 6.2.3. Cells of Myeloid Origin: Tumour-Associated Macrophages (TAM), Tumour-Associated Neutrophils (TAN), and Myeloid-Derived Suppressor Cells (MDSC)

They originate from the common myeloid progenitor [50]. TAM and TAN can be antitumour (type 1) or pro-tumour (type 2). In ccRCC, they are mainly pro-tumour (type 2) [51]. MDSCs play an important role in the escape of tumour cells from the immune system. They are in an immature state and can suppress T-cell responses. There are two subpopulations of MDSCs: (i) monocytic MDSCs (M-MDSCs) decrease the amount of available L-arginine and thus inhibit the formation of a functional TCR and the proliferation of T-lymphocytes; (ii) granulocytic MDSCs (G-MDSCs) suppress the response of CD8+T-lymphocytes by producing reactive oxygen species (ROS). Like TANs and TAMs, MDSCs inhibit the innate immune system, promoting tumour angiogenesis and metastasis. The primary target immune cell population inhibited by MDSCs are T cells. MDSCs primarily target and inhibit T cells through various mechanisms such as activation and proliferation inhibition, induction of anergy, apoptosis-driven T cell depletion, and homing blockade. These mechanisms, which have been discussed in detail in recent articles [52,53,54,55], can be divided into five categories: (i) secretion of immunosuppressive molecules (e.g., IL-6, IL-10, TGF-β, ROS, NOS, PD-1, PD-L1, CTLA-4, VEGF); (ii) degradation of metabolites critical for T cell functions; (iii) manipulation of chemotactic molecules that control T cell homing; (iv) induction of immunosuppressive cells, such as T regulatory (Treg) cells; and (v) alteration of adenosine metabolism by expression of ectoenzymes. Therefore, MDSCs appear to be an important target for overcoming resistance to immune checkpoint inhibitor therapy [56], and targeting IL6 with existing therapeutic antibodies may, therefore, improve the efficacy of immune checkpoint inhibitor therapy [57]. Finally, all these cells express PDL1 on their surface. Resistance to immunotherapy has been associated in preclinical models of colorectal and breast cancer with an accumulation of circulating MDSC as well as with the presence of TAM, TAN, and MDSC in the tumour [58]. ELR+CXCL produced by tumour cells bind to CXCR1/2 expressed by TAM, TAN, and MDSC and thus promote their activation and retention in the tumour [59]. Cells of myeloid origin also secrete ERL+CXCL, which promotes autocrine activation and attraction. Targeting cells of myeloid origin is a promising therapeutic approach to overcome immunosuppression and increase the efficacy of immunotherapy [60].

#### 6.2.4. Cancer-Associated Fibroblasts (CAF)

TGFβ, PDGF, and the various interleukins (IL6, ELR+CXCL, etc.) induce the differentiation of tumour fibroblasts into CAF. CAF produce numerous cytokines (IL6, IL8, IL10, TNFα, TGFβ, etc.) and promote the attraction of cells of myeloid origin (TAM, TAN, MDSC) and the differentiation of myeloid cells into MDSC in the TME [61]. In addition, CAF promote the attraction of T lymphocytes and stimulate their differentiation [62,63]. Studies have shown that CAF-derived factors such as TGFβ and IL6 can influence the balance between Treg (inhibition of immune responses) and T helper 17 (Th17) (autoimmunity and inflammation). For example, TGFβ is known to promote Treg differentiation, while IL6 can inhibit Treg differentiation and promote Th17 differentiation [64,65]. They also generate a collagen matrix that prevents T cells from invading the tumour. The presence of CAF has been associated with insensitivity to immunotherapy in preclinical models of colorectal and pancreatic cancer [66]. CAF express CXCR2 and produce substantial amounts of ERL+CXCL. This production has been linked to the attraction of cells of myeloid origin in tumours involved in the establishment of immune tolerance [66,67].

## 7. Combination of Anti-Angiogenics and Immunotherapies: The Holy Grail for Clear Cell Renal Cell Carcinoma?

### 7.1. Unlocking the Potential: The Rationale for Immunotherapies in Clear Cell Renal Cell Carcinoma

The development of immunotherapies has revolutionised the therapeutic landscape for various types of cancer, particularly ccRCC. As mentioned earlier, anti-angiogenic drugs were the first treatments to extend progression-free survival by months or years in patients whose prognosis was very poor before 2007 and the approval of anti-VEGF antibodies and pharmacological inhibitors of tyrosine kinase receptors. 

Immune checkpoint inhibitor therapy was first used in the second line after a relapse on anti-angiogenics. A phase III clinical trial showed improved overall survival and fewer grade 3 or 4 adverse events compared to everolimus [68]. At the molecular level, this almost empirical administration of immune checkpoint inhibitor therapy was rationalised by the immunological detection of PDL1 in tumour samples from patients treated with either sunitinib or bevacizumab. Infiltration of Tregs in these tumours was also increased when resistance to anti-angiogenic treatments occurs [69,70,71]. This landmark publication demonstrated that anti-angiogenic therapies inhibit tumour vascularisation but form an immune-tolerant landscape that leads to further tumour progression.

Another pivotal discovery made by Voron et al. underscores that VEGFA, generated within the tumour microenvironment, activates VEGF receptor 2 on T cell surfaces. This activation, in turn, triggers the expression of various immune checkpoints, including CTLA4, PD1, LAG3, and TIM3 [72]. In experimental tumour settings where anti-angiogenic drugs, including TKIs or anti-VEGF antibodies, demonstrated efficacy, a notable outcome emerged: there was a discernible reduction in the expression of immune checkpoints on the surface of T cells. This intriguing finding suggests a potential interplay between the effectiveness of anti-angiogenic therapies and the modulation of immune regulatory mechanisms. The observed decrease in immune checkpoint expression highlights a dynamic connection between angiogenesis inhibition and immune response regulation. Therefore, anti-angiogenic therapies inhibit tumour neo-vascularisation but also prevent immune tolerance (Figure 3).

Finally, a very important concern is driving immune cells to the tumour. The original hypothesis was that inhibiting the development of blood or lymphatic vessels should be a fallacy. However, several publications suggest that anti-angiogenic drugs do not destroy blood or lymph vessels but normalise them to promote better blood flow or lymph drainage. This theory, put forward by R. Jain, partly explains the relatively low efficacy of anti-angiogenic drugs as the sole therapy [73]. For this reason, the anti-VEGF antibodies bevacizumab/Avastin have been combined with conventional therapy in the pivotal clinical trials for colorectal cancer [74], lung cancer [75], breast cancer [76], renal cell carcinoma [77], and ovarian cancer [78]. The same principle is applied to the combination of anti-angiogenic and immunotherapies. 

### 7.2. Optimising Therapeutic Synergies: Strategic Combinations and Future Trajectories

Given the great impact of combinations of immunotherapies in melanoma patients, the same questions arise for ccRCC. Which was the best therapeutic regimen: a combination of immunotherapies or a combination of anti-angiogenic with immunotherapies?

Therefore, several clinical trials were conducted in ccRCC to evaluate the different combinations. The first trial looked at the combination of immunotherapies used in melanoma, ipilimumab (anti-CTLA4), and nivolumab (anti-PD1) [79]. This combination shows better outcomes in overall survival and objective response rate than the current standard treatment with sunitinib. Then, several combinations of anti-angiogenic drugs and ICI were studied, including bevacizumab (anti-VEGF)/atezolizumab (anti-PDL1) [80], axitinib/pembrolizumab (anti-PD1) [81], and axitinib/avelumab (anti-PDL1) [82]. Of these three combinations, axitinib/pembrolizumab proved to be the best and became the standard first-line treatment. More recently, two other combinations of anti-angiogenic therapies with ICI have achieved impressive results. Cabozantinib/nivolumab [83] and lenvatinib/pembrolizumab [84] outperformed sunitinib in phase III clinical trials.

But in this rapidly changing landscape, what is the best strategy? A seminal piece of work has been the discovery of genetic subgroups that are better suited to anti-angiogenic therapies or immunotherapies [85] or alternative treatments [86]. A clinical trial highlights the importance of specific biological markers in predicting patients’ suitability for such treatments [87]. Despite these major improvements, ccRCC is still not curable.

Undoubtedly, further studies are imperative to elucidate this pivotal question and uncover additional mechanisms that contribute to patient relapse. The pursuit of a comprehensive understanding in this area is essential for improving treatment strategies and enhancing our capability to deal with the complexities linked to patient outcomes. The quest for deeper insights into the factors influencing relapse will pave the way for more effective and personalised interventions. In this context, numerous studies have highlighted the connection between immunotherapy resistance or relapse and the presence of fibrosis. The tumour microenvironment (TME) emerges as a key player in the development of acquired resistance to immunotherapy, with fibrosis constituting a notable and influential component within the TME.

## 8. Unlocking the Therapeutic Synergy: Integrating Anti-Angiogenics, Immunotherapies, and Anti-Fibrosis in the Pursuit of Optimal Management for Clear Cell Renal Cell Carcinoma

### 8.1. Fibrosis, Resembling a Powerful Catalyst, Actively Fuels and Propels the Growth of Tumours

Renal fibrosis emerges because of chronic kidney disease (CKD), presenting an unresolved medical challenge that represents an unmet medical need.

CKD is a progressive condition that affects >10% of the general population worldwide, amounting to >800 million individuals [88]. Despite several trials, CKD remains incurable without real improvement with anti-TGFβ, pirfenidone, galectin antagonist, and antibodies against αvβ6 integrin [89]. All fibrotic diseases, including CKD, are characterised by the progressive accumulation of extracellular matrix (ECM) components, leading to organ failure. Fibrosis, a consequence of the chronic inflammatory process, creates a conducive environment for tumour development, emphasising its pro-tumourigenic properties. Understanding and targeting the mechanisms behind fibrosis is crucial for devising strategies that impede its supportive role in tumour progression.

Fibrosis is a common feature frequently observed in ccRCC. Intratumoural fibrosis is a histologic manifestation of fibrotic tumour stroma, and its presence in ccRCC has been associated with poor prognosis and cancer aggressiveness. Intratumoural fibrosis is positively correlated with the histological grade of ccRCC and intratumoural inflammation [90,91]. The interaction between cancer cells and fibrotic stroma is intricate and reciprocal, involving dysregulations from multiple biological processes.

### 8.2. Fibrosis: A Key Contributor to Immunotherapy Resistance?

While the limited success of immunotherapy in cancer treatment has traditionally been attributed to intrinsic tumour characteristics, such as low immunogenicity, a diminished mutational burden, and a compromised host immune system, recent analyses of clinical trial results across various cancer types reveal a common thread: tumours with substantial fibrotic stroma often exhibit suboptimal or negligible responses to treatments [92]. 

The TME is predominantly composed of a fibrotic stroma, primarily composed of connective tissue. CAFs actively drive significant remodelling in this stroma, contributing to the accumulation of an excessive ECM. Both elements of the fibrotic connective tissue—the cellular component dominated by CAFs, and the non-cellular aspect characterised by a rigid ECM—contribute to the establishment of an immunosuppressive microenvironment and serve as a physical impediment to effective drug infiltration. Consequently, this dual impact decreases the efficacy of anti-tumour immunotherapies.

### 8.3. Inflammatory Environments and Fibrosis: Unraveling the Complex Interplay

When epithelial and endothelial cells become compromised due to either treatment or chronic inflammation, they release chemotactic factors, initiating the recruitment of inflammatory macrophages and neutrophils to the affected site. These recruited cells subsequently release reactive oxygen species (ROS), cytokines, and chemokines, activating mesenchymal cells. This activation leads to the production of ECM and further amplifies the synthesis of pro-inflammatory cytokines and angiogenic factors, creating an environment conducive to the development of cancerous lesions.

Furthermore, the accumulation of senescent cells during ageing, along with the release of various senescence-associated secretory phenotypes (SASP), plays a crucial role in the initiation of fibrosis and tumourigenesis. Signalling cascades triggered by TGFβ, involving cytokines and pro-fibrotic factors, emerge as significant therapeutic targets. Despite this, specific therapies capable of halting or reversing fibrosis remain elusive. Epidemiological studies have established a clear link between fibrosis, cancer, and resistance to therapy [93]. 

In the context of cancer aggressiveness, ELR+CXCL chemokines (CXCL1, 2, 3, 5, 6, 7, 8) and their receptors CXCR1/2 play a pivotal role [94]. Their activation stimulates fibrosis by initiating mesenchymal cell and neutrophil migration. In ccRCC, the ELR+CXCL/CXCR1/2 axis promotes tumour cell proliferation and angiogenesis, mirroring observations of CKD with a similar endpoint of chronic inflammation and the development of cancer-associated fibroblasts. These effects are further induced by TKIs targeting the VEGF/VEGFR pathway and contribute to resistance against anti-angiogenic drugs designed to target VEGF and its receptors [44]. Additionally, VEGFC, a significant driver of lymphangiogenesis produced by tumour cells and M2 macrophages, not only induces resistance to anti-angiogenics but also plays a crucial role in fibrosis through crosstalk with TGFβ [95]. ELR+CXCL further induce lymphangiogenesis via VEGFC [96]. Genetic disruption of VEGFC or its targeted inhibition has been demonstrated to inhibit the growth of anti-VEGF/VEGFR-resistant ccRCC [37,97]. VEGFC’s involvement in driving fibrosis through crosstalk with TGFβ underscores its pivotal role [95].

The stiffness of the matrix is determined by a balance between ECM degradation and build-up. During tumour progression, different cell types (tumour cells, macrophages, fibroblasts, etc.) continuously produce matrix-crosslinking enzymes such as matrix metalloproteinases (MMPs) and tissue inhibitors of metalloproteinases (TIMPs) to promote ECM remodelling. In humans, 24 MMPs are currently known, classified according to their preferred substrates, including collagenases (MMP1, 8, 13) and gelatinases (MMP2, MMP9) [98]. TIMPs serve as natural inhibitors of MMPs and traditionally hinder ECM degradation [99]. In ccRCC, the balance between MMPs and TIMPs is disturbed and shifted towards MMPs [100]. Serum levels of MMP2 and MMP9 have been found to be significantly elevated in tumour tissue and urine samples from ccRCC patients [101,102]. Elevated serum levels of MMP2 and 9 have also been found in CKD patients [103]. Numerous studies indicate a detrimental role of MMPs in both early and later stages of kidney disease, particularly with their ECM-degrading properties during scarring and fibrosis [104,105]. Peptides upregulated in renal pathologies, such as angiotensin-II, contribute to renal fibrosis by influencing MMP expression, thereby affecting crucial remodelling processes [106,107]. MMPs play a multifaceted role in kidney disease, involving cell migration, cell–cell and cell–matrix adhesion, as well as the release and activation of extracellular matrix-bound growth factors and cytokines. Several of these functions have been implicated in the initiation and progression of CKD and kidney fibrosis [108].

## 9. Connective Tissue Growth Factor (CTGF)—The Essential Link between VEGFC, Lymphangiogenesis, and Fibrosis

### 9.1. CTGF and Its Multifaceted Role

Factors induced by TKI treatment contribute to the establishment of a persistent inflammatory environment that promotes and sustains the fibrotic state. Such a link between fibrosis and TKI treatment has been elucidated in the context of lung disease [109]. Identifying the key factors involved in ccRCC fibrosis is essential to prevent relapse and treatment failure.

As previously mentioned, sunitinib exposure induces an inflammatory secretome. Among the different cytokines produced, CTGF (connective tissue growth factor) is highly induced in resistant cells to sunitinib [32]. CTGF, a secreted protein belonging to the CCN family (CYR61 (cysteine-rich61)/CTGF (connective tissue growth factor)/N OV (nephroblastoma overexpressed)), stands as the second member, also recognised as CCN2. Across the CCN family, shared molecular structures include the IGF binding domain, von Willebrand factor C (VWC) domain, thrombospondin type 1 domain, and the C-terminal domain. Notably, the TSP domain and C-terminal domain engage with diverse integrins, such as integrin α6β1, αvβ3, or α5β1, orchestrating key signalling pathways such as MAPK, WNT, NFκB, and ERK. The VWC domain plays a pivotal role in fibrosis and apoptosis, driven by its interaction with TGFβ. Creating a complex web of interactions, CTGF establishes a positive feedback loop with various cytokines like TGFβ, VEGF, and integrins, leading to diverse fibrotic diseases. CTGF, a versatile growth factor, is implicated in various biological processes such as wound healing, inflammation, cell adhesion, chemotaxis, apoptosis, angiogenesis/lymphangiogenesis, and fibrosis (Figure 4). Its impact extends to the proliferation of tumour cells, where increasing evidence highlights CTGF’s involvement in cancer initiation, progression, and metastasis. By driving cell proliferation, migration, invasion, drug resistance, and epithelial–mesenchymal transition (EMT), CTGF emerges as a key player in the complex landscape of cancer biology. Moreover, CTGF’s influence extends to the tumour microenvironment, contributing to angiogenesis, inflammation, and the activation of CAF across various nodal sites. Identified as an oncogene in multiple cancers, including melanoma [110], chondrosarcoma [111], acute lymphoblastic leukaemia [112], and pancreatic cancer [113], CTGF’s clinical relevance is underscored by its positive correlation with bone metastasis [114], glioblastoma growth [115], poor prognosis in oesophageal adenocarcinoma [116], aggressive behaviour of pancreatic cancer cells [117], and invasive melanoma [118]. Clinically, CTGF expression emerges as a reliable indicator of progression, poor prognosis, and metastasis across diverse cancers. Its role in promoting chemoresistance in breast [119] and ovarian cancers [120] adds another layer to its significance. Moreover, mice lacking the CTGF gene exhibit defects in angiogenesis and vascular integrity, underscoring the pivotal role of CTGF in these processes [121]. Conversely, transgenic mice overexpressing CTGF display enhanced renal fibrosis following events such as unilateral ureteral obstruction.

The multifaceted landscape of CTGF is intricately depicted. CTGF expression is responsive to diverse stimuli, including stress, growth factors (e.g., TGFβ), and transcription factors. Interacting with a spectrum of entities such as cell surface receptors (integrins, HSPGs, EGFR), extracellular matrix proteins (fibronectin, aggrecan), and cytokines (IL1, IL6), CTGF emerges as a pivotal player in various physiological processes. It contributes significantly to cell adhesion, extracellular matrix production, tissue wound repair, and angiogenesis. However, the dual nature of CTGF becomes evident in its involvement in pathological contexts. It is implicated in inflammation, fibrosis (e.g., lung, kidney), abnormal angiogenesis, and various cancers, including breast and lung cancer. This comprehensive overview highlights the diverse roles and contextual implications of CTGF in health and disease.

### 9.2. CTGF, Fibrosis, and Therapeutic Implications

The interaction between intratumour fibrosis and tumours, promoting growth and spread, is now well-established. Fibrosis induces metabolic changes in tumours and surrounding microenvironmental cells, alters cell adhesion through the expression of fibronectin and collagen, and through the balance of MMPs and TIMPs, which are involved in the degradation of the ECM components. Fibrosis also facilitates immune escape by diminishing T lymphocyte interaction with tumour cells. The pro-fibrotic role of CTGF was first demonstrated in 1999 by Mori and colleagues [122], who showed that subcutaneous injection of CTGF and TGFβ induced long-term fibrotic tissue. Furthermore, studies indicate that CTGF is highly expressed in systemic sclerosis, a disease characterised by severe fibrosis affecting various organs such as the skin, digestive tract, lung, and heart. The use of a monoclonal antibody specific for CTGF attenuates skin fibrosis in a murine model of systemic sclerosis [123]. In cancer, CTGF has been associated with poor prognosis in mesothelioma, particularly when expressed by CAFs. A CTGF-specific monoclonal antibody was effective for mesothelioma in a murine model [124]. Finally, in the context of chronic kidney conditions in humans, CTGF plays a crucial role in promoting the extension of tissue fibrosis, as evidenced by its strong correlation with cellular proliferation and matrix accumulation.

Lymphatic vessels play a central role in fibrotic diseases. VEGFC, VEGFD, and VEGFR3 are key molecules involved in lymphangiogenesis. Human kidney biopsies have shown an increase in lymphatic vessels in fibrotic territories, particularly in CKD [125]. CTGF trigger the production of VEGFC, essential for fibrosis and metastatic dissemination following resistance to anti-angiogenic drugs [34]. VEGFC interacts with the full-length form of CTGF, which limits its ability to stimulate lymphangiogenesis [126]. These two antagonistic results probably indicate negative retrocontrol. Therefore, metalloproteinases associated with the tumour environment by maturating CTGF liberate VEGFC to stimulate lymphangiogenesis [126]. CTGF knockdown led to a decrease in VEGFC expression and lymphangiogenesis, accompanied by a significant reduction in fibrosis in a model of obstructive nephropathy [126]. Furthermore, in a model of peritoneal fibrosis, the number of lymphatic vessels and VEGFC increased, and this effect was reversed using a TGFβ inhibitor [127]. Additionally, TGFβ can induce the expression of CTGF in mesangial cells and renal tubular epithelial cells [128]. CTGF appears to enhance lymphangiogenesis through the phosphorylation of ERK, depending on the interaction between αvβ5 and αvβ3 [129]. 

MMPs and TIMPs emerge as pivotal contributors to the pathogenesis of fibrosis. The overexpression of CTGF increases MMP2 and MMP3 expression and promotes cell migration [130,131]. Notably, MMP1 and MMP13 cleave full-length CTGF into N-terminal and C-terminal fragments, both possessing similar molecular weights [132,133]. The cleavage of full-length CTGF into these fragments demonstrates enhanced potency in inducing hepatic fibrogenesis. Specifically, the N-terminal fragment of CTGF mediates myofibroblast differentiation and collagen synthesis and functions as a downstream mediator of TGFβ [134,135]. Conversely, the C-terminal fragment is implicated in stimulating fibroblast proliferation [135]. This intricate interplay highlights the diverse roles of CTGF and its processing by MMPs in the complex cascade of events leading to fibrosis.

CTGF full length and N and C-terminal fragments play a significant role in fibrosis. It is a central mediator of tissue remodelling and fibrosis, and its inhibition can reverse the process of fibrosis. However, the role of CTGF and the therapeutic efficacy of targeting CTGF in renal cancer progression and metastasis are still unknown and require further investigation. In ccRCC, CTGF has been found to be overexpressed in tumour tissues [136]. Therefore, targeting CTGF may represent a promising strategy in cancer, particularly in ccRCC, where intratumoural fibrosis might lead to cancer aggressiveness and is related to poor prognostic parameters, including Fuhrman nuclear grade, intratumour necrosis, and lymphovascular invasion.

## 10. Conclusions

In general, the overall fibrotic response plays an important role, both directly and indirectly, in compromising the efficacy of treatment and immunotherapy. The interactions between the elements of a fibrotic response, the tumour cells, and the immune cells are complicated and highly interconnected. Ultimately, the question of whether tumour-associated fibrosis impedes the activity of cytotoxic immune cells and/or enhances the activity of immunosuppressive regulatory immune cells will determine the success or failure of immunotherapy. Therefore, many ongoing studies focus either on fibrosis itself or on signalling molecules that promote fibrosis. One particularly well-researched signalling molecule with significant therapeutic potential is TGFβ. Since TGFβ signalling contributes to the maintenance of fibrotic responses and hyperactivation of CAFs and can directly impact immune cell functions, the development of anti-TGFβ antibodies and/or TGFβ inhibitors in combination with conventional immunotherapies is currently being explored. However, it is important to note that blocking TGFβ may have conflicting effects on clinical outcomes, as it is a pleiotropic molecule that has both tumour-promoting and tumour-inhibiting functions. Other strategies target structural elements of the ECM, such as collagen or collagen cross-linking enzymes, fibronectins, etc., or cytokines, chemokines, or other non-structural proteins, such as members of the CCN family, like CTGF [137]. Targeting CTGF may represent a promising strategy in cancer, particularly in clear-cell renal cell carcinoma (Figure 5).

## 11. Future Directions

The intricate role of CTGF in the complex interplay between fibrosis, pre-tumoural conditions, and the emergence of aggressive metastatic diseases remains a subject of intense investigation. The question of whether CTGF serves as the primary driver of metastasis or is a consequence of VEGFC overexpression, a pivotal mediator of lymphangiogenesis, underscores the need for deeper exploration.

Despite initial optimism, recent findings from clinical trials, such as NCT04371666 for Duchenne muscular dystrophy and Phase III trials (NCT04419558, NCT03955146) targeting Idiopathic Pulmonary Fibrosis (IPF), have tempered expectations. The anti-CTGF therapy using the humanised antibody pamrevlumab showed limited efficacy in inhibiting fibrosis, even in well-tolerated treatments. Ongoing research, exemplified by the current phase III clinical trial (NCT03941093) focusing on metastatic pancreatic cancer with results expected in 2024, further highlights the challenges associated with harnessing CTGF as a transformative treatment for highly fibrotic disorders.

In light of these findings, it is clear that the development of CTGF-based therapies requires a more nuanced and comprehensive understanding of their mechanisms and potential limitations. Moving forward, a concerted effort in research and development is essential to refine existing theories and establish a solid foundation for the potential clinical application of CTGF-targeted treatments in highly fibrotic disorders.

## Figures and Tables

**Figure 1 biomedicines-12-00385-f001:**
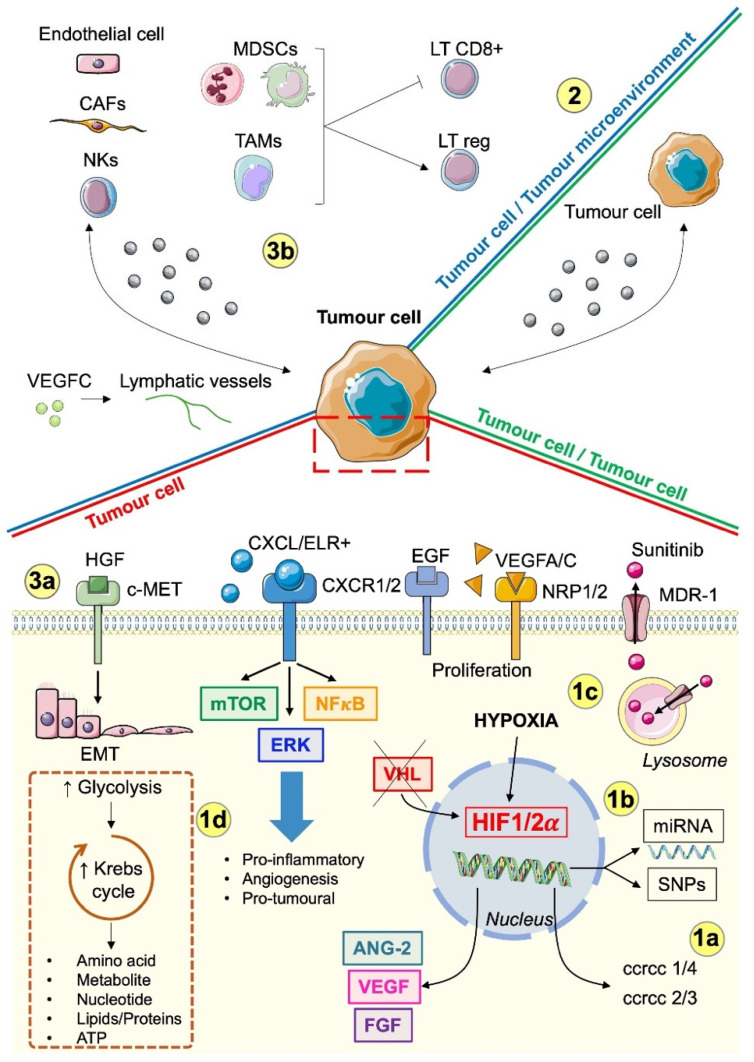
Summary of mechanisms of resistance to anti-angiogenic drugs in mRCC.

**Figure 2 biomedicines-12-00385-f002:**
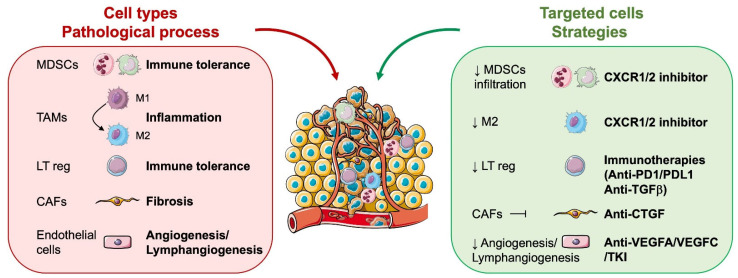
The importance of combining anti-angiogenic drugs and immunotherapies to inhibit the crosstalk between tumour cells and immune cells.

**Figure 3 biomedicines-12-00385-f003:**
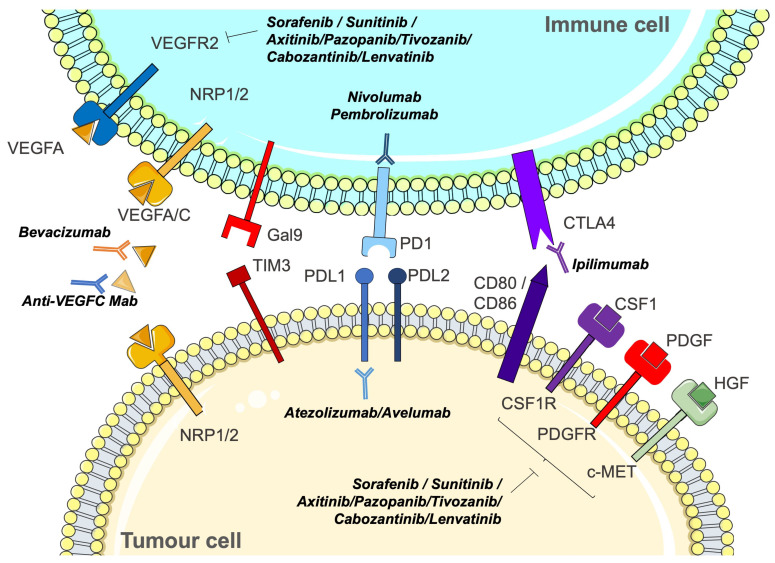
Cells involved in the pathological process in the tumour microenvironment and the strategies to combat them. Cells of the tumour microenvironment, including MDSCc, TAM, M1 and M2, llymphocytes T reg, and CAFs. The different solutions considered are presented, including CXCR2 inhibitors, immunotherapies, such as PD1, and anti-PDL1 and anti-PDL2.

**Figure 4 biomedicines-12-00385-f004:**
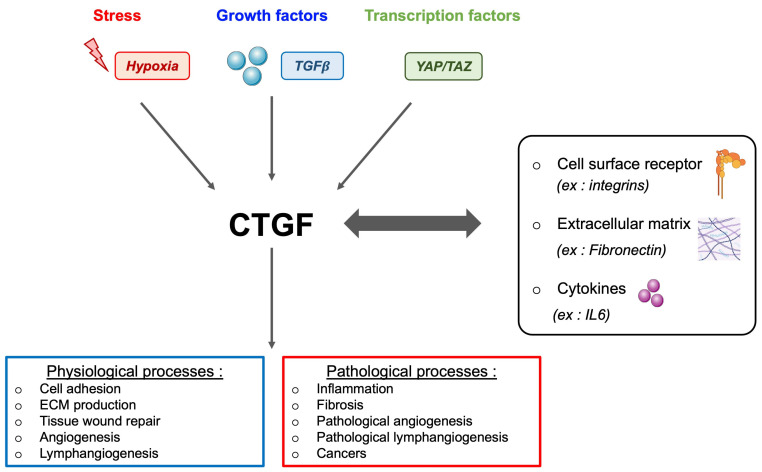
Unveiling the dynamics of CTGF: induction, interactions, and impact overview.

**Figure 5 biomedicines-12-00385-f005:**
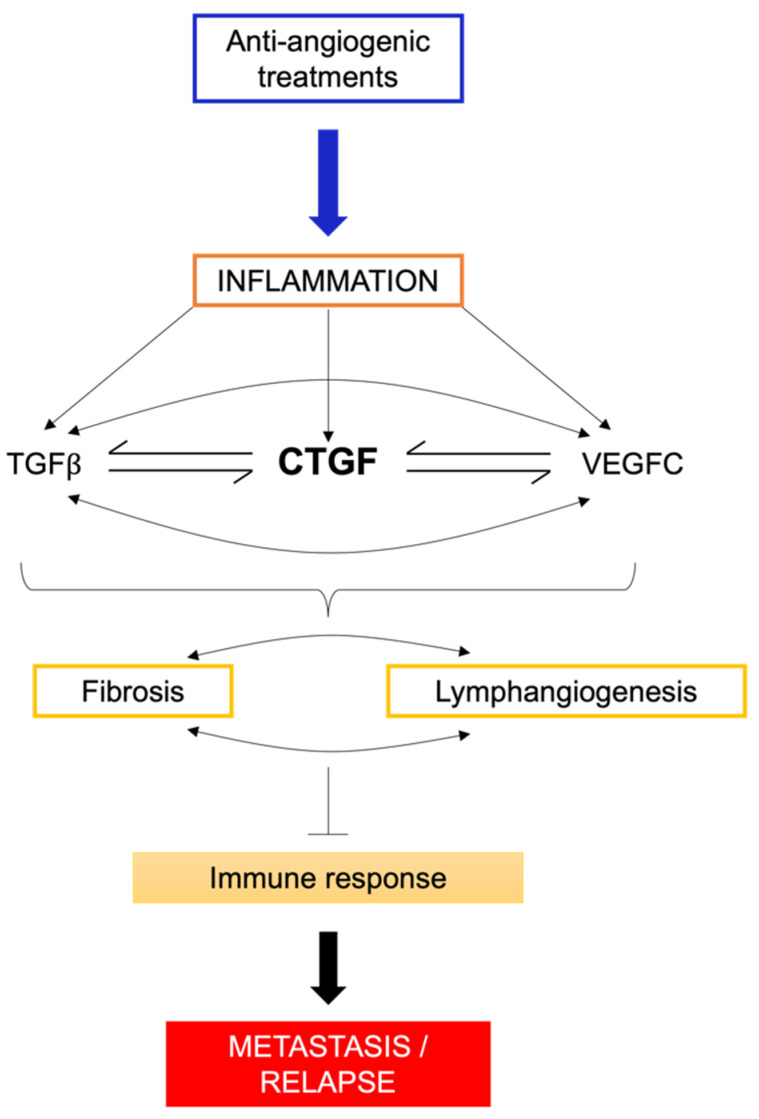
Summary schematic of the relationship between mechanisms of resistance to anti-angiogenic treatments and tumour recurrence, considering CTGF as an important mechanism of fibrosis (a precancerous condition) and lymphangiogenesis, a key phenomenon in metastatic spread.

## Data Availability

Not applicable.

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
