# Peer review of "Combination of Anti-Angiogenics and Immunotherapies in Renal Cell Carcinoma Show Their Limits: Targeting Fibrosis to Break through the Glass Ceiling?"

_biomedicines, 2024, doi:10.3390/biomedicines12020385_

Round 1

Reviewer 1 Report

Comments and Suggestions for Authors

This review explores treatments for metastatic clear cell renal cell carcinoma, examining antiangiogenic therapies and immunotherapies, and emphasizes their limitations. The analysis focuses on fibrosis, particularly the role of connective tissue growth factor (CTGF), suggesting its targeting as a promising approach to address challenges in ccRCC, while underscoring the need for further research to integrate CTGF-targeted therapies effectively.

Comments:

1. The subsections and the details covered are adequate and detailed.

2. The paper presents the literature and concepts clearly.

3. The paper addresses an intriguing topic and presents valuable insights into multiple approaches.

4. The paper is well-written, and the results are promising.

5. Special appreciation for formatting good quality figures.

Overall this is a thoroughly researched, expertly written and carefully segmented presentation. This paper can be accepted as is. Congratulations to the authors.

Author Response

We thank the reviewer for the kind remarks.

Reviewer 2 Report

Comments and Suggestions for Authors

Overall, the article has the hallmark of being written by one of the artificial intelligence programs, especially ChatGPT. While there is nothing wrong with keeping up with technology, it is the responsibility of the authors to ensure that it scientifically accurate. The article has linguistically correct, but scientifically meaningless sentences. This is not just in only one place but throughout the article and therefore, it is impossible to comment on every sentence. Some examples are given below.

Reference 6 is not a combination study with bevacizumab and interferon alpha. It is a comparison with sunitinib and interferon alpha.

Lines 43-67 do not read like a scientific document. Non-responsiveness and drug resistance are two distinct features. This needs to be delineated, and resistance should be discussed in terms of intrinsic and acquired resistance with a scientific tone.

Line 74: ‘On the flip side” – what does this mean?

Lines 81-87, needs to be re-written with a scientific perspective.

Lines 111-116, as above

Line 120 – what is the elegant hypothesis? That anti-angiogenic agents indeed  inhibit angiogenesis?

Line 382, the central theme of the article: “Renal fibrosis is a consequence of chronic kidney disease (CKD) which represents an unmet medical need”. This sentence is scientifically meaningless.

Author Response

Point by point answers to Reviewer 2

While there is nothing wrong with keeping up with technology, it is the responsibility of the authors to ensure that it scientifically accurate. The article has linguistically correct, but scientifically meaningless sentences. This is not just in only one place but throughout the article and therefore, it is impossible to comment on every sentence. Some examples are given below.

We have done our best to correct the wording of the text so that it is scientifically meaningful.

Reference 6 is not a combination study with bevacizumab and interferon alpha. It is a comparison with sunitinib and interferon alpha.

We apologize if this was not clear enough. Reference 5 corresponds to the historical use of the combination of bevacizumab plus interferon as first line and reference 6 corresponds to a different treatment comparing sunitinib with the then reference treatment interferon. We have amended the sentence as follows: “Therefore, ccRCC represents a paradigm for treatment with anti-angiogenic drugs [4], including the historical combination of bevacizumab/avastin (humanised monoclonal antibody) with interferon alpha [5] and the use of the multitarget inhibitor sunitinib [6]”

Lines 43-67 do not read like a scientific document. Non-responsiveness and drug resistance are two distinct features. This needs to be delineated, and resistance should be discussed in terms of intrinsic and acquired resistance with a scientific tone.

This part is an introductory text, and the mechanisms are discussed afterwards. We changed the text to the following: “This diversity of patient outcomes underscores the multifaceted nature of resistance, which is due to factors such as genetic or epigenetic aberrations in cancer cells, environmental influences, and interactions with host cells. These mechanisms can either manifest themselves before the start of therapy or occur later. Each patient has a unique genetic profile, which requires a personalized therapeutic approach. However, achieving this goal requires a comprehensive understanding of the specific genetic landscape and the underlying mechanisms that determine adaptation to treatments with varying efficacy. Resistances identified in cancer therapies can be divided into intrinsic and acquired forms depending on the temporal context of its manifestation. This review examines in detail the temporal dimensions that influence resistance and provides insights into the different mechanisms underlying this phenomenon (see Figure 1). In addition, possible ways of deciphering the factors contributing to treatment failure are outlined.”

Line 74: ‘On the flip side” – what does this mean?

It has been deleted.

Lines 81-87, needs to be re-written with a scientific perspective.

We changed the text to; “Germline genetic alterations in the promoter region of different genes have been shown to be clinically relevant and are potential predictive markers for the efficacy of antiangiogenic therapies. SNPs, which are characterized by subtle genetic changes in which a single nucleotide is exchanged, exert a significant influence. SNPs can predict response to treatment, susceptibility to environmental factors and predisposition to certain diseases. In ccRCC, SNPs play a central role as they influence both development and resistance to anti-angiogenic therapies. The VEGF gene, for example, is highly polymorphic and many single nucleotide polymorphisms (SNPs) have been described [14]. Also, a compelling illustration of this phenomenon is found in the intriguing correlation between specific IL-8 SNPs and reduced survival rates under sunitinib treatment. This correlation is a consequence of the activation of alternative angiogenic pathways, underscoring the profound implications of SNPs in shaping the dynamics of response to therapeutic interventions [14]”.

Lines 111-116, as above

The editor has suggested additional mechanisms linked to miR.

The text was changed to: “miR-15b was found to be significantly upregulated in cell lines resistant to sunitinib after exposure to the drug. This observation has been replicated in in vivo models [17]. Other notable miRNAs overexpressed in sunitinib-resistant RCC cells include miRNA-575, miRNA-642b-3p, and miRNA-4430 (in vitro study), as well as miRNA-942, miRNA-133a, miRNA-628-5p, and miRNA-484 (in vivo study) [16,18]. Overexpression of miR-144-3p in ccRCC was shown to increase cell proliferation, clonogenicity, migration, invasion, and resistance by inhibiting the expression of the tumor suppressor gene ARID1A which encodes the basic directional subunit of SWI/SNF chromatin remodeling complexes [19]. In addition, miR-221, attenuates the efficacy of sunitinib by reducing VEGFR2 expression. Additionally, miR-942 up-regulates MMP-9 and VEGF secretion which promote endothelial cell migration and sunitinib resistance [16]. In contrast, miR-200b and miR-141 were found to be downregulated in ccRCC compared to benign tissue, and their expression may serve as an independent prognostic factor for prolonged progression-free survival and overall survival [20].”

Line 120 – what is the elegant hypothesis? That anti-angiogenic agents indeed inhibit angiogenesis?

Yes, that was the original hypothesis: only the inhibition of blood vessels that cannot adapt to treatments because they have a reduced genetic adaptability. However, antiangiogenic agents also act on tumor cells, as discussed below, and thus induce resistance mechanisms in tumor cells.

Line 382, the central theme of the article: “Renal fibrosis is a consequence of chronic kidney disease (CKD) which represents an unmet medical need”. This sentence is scientifically meaningless.

Kidney fibrosis is an important medical challenge. We changed the text to “Renal fibrosis emerges because of chronic kidney disease (CKD), presenting an an unresolved medical challenge which represents an unmet medical need.

Reviewer 3 Report

Comments and Suggestions for Authors

This study suggests that targeting fibrosis as a treatment for RCC could lead to the development of new therapeutic strategies. Overall, this paper is interesting. The reviewer thinks that this paper is useful information for readers. However, the reviewer would like to suggest some critiques as follows.

1.     From line 18 to line 21, these sentences appear to be correct, however, the reader may find the wording confusing. It would be better to describe it more concisely.

2.     The authors use the term "kidney cancer," suggesting that the term "renal cell carcinoma" should be used instead.

3.     On line 48 and 53, " inevitably relapse or progression" seems to be an exaggeration.

4.     On line 50, the authors should use “progression” instead of “relapse.”

5.     Authors should describe the search strategy and selection criteria for literature selection.

Author Response

This study suggests that targeting fibrosis as a treatment for RCC could lead to the development of new therapeutic strategies. Overall, this paper is interesting. The reviewer thinks that this paper is useful information for readers. However, the reviewer would like to suggest some critiques as follows.

We thank the reviewer for constructive comments.

  1. From line 18 to line 21, these sentences appear to be correct, however, the reader may find the wording confusing. It would be better to describe it more concisely.

We changed the text to the following: “We discuss the potential of targeting CTGF to address the problem of fibrosis in ccRCC. Emphasizing the crucial relationship between fibrosis and the immune system in ccRCC, we propose that targeting CTGF holds promise for overcoming obstacles to cancer treatment. However, we recognize that an in-depth understanding of the mechanisms and potential limitations is imperative and therefore advocate for further research. This is an essential prerequisite for the successful integration of CTGF-targeted therapies into the clinical landscape.” We hope it will be clearer.

  1. The authors use the term "kidney cancer," suggesting that the term "renal cell carcinoma" should be used instead.

We corrected accordingly.

  1. On line 48 and 53, " inevitably relapse or progression" seems to be an exaggeration.

We changed the text to the following: “Despite their relative effectiveness, these treatments are not a curative solution and patients progress after varying lengths of time. The heterogeneous responses of patients can be divided into three distinct categories: 1) patients with intrinsic resistance who exhibit no or minimal response, facing rapid progression and mortality; 2) patients who have acquired resistance and experience transient benefits followed by relapse; 3) and a subgroup for whom treatment proves effective over an extended timeframe [12].”

  1. On line 50, the authors should use “progression” instead of “relapse.”

We changed it accordingly.

  1. Authors should describe the search strategy and selection criteria for literature selection.

The team's expertise in the mechanism of resistance to antiangiogenic drugs facilitated the selection of relevant articles for the literature review. We have been collecting this literature for more than 10 years as we are at the forefront of the progress of the various mechanisms highlighted in this review.